# A Novel lncRNA Mediates the Delayed Tooth Eruption of Cleidocranial Dysplasia

**DOI:** 10.3390/cells11172729

**Published:** 2022-09-01

**Authors:** Yuejiao Xin, Yang Liu, Jie Li, Dandan Liu, Chenying Zhang, Yixiang Wang, Shuguo Zheng

**Affiliations:** 1Department of Preventive Dentistry, Peking University School and Hospital of Stomatology, National Center of Stomatology and National Clinical Research Center for Oral Diseases, National Engineering Research Center of Oral Biomaterials and Digital Medical Devices, Beijing 100081, China; 2Department of Periodontics, Tianjin Stomatological Hospital, School of Medicine, Nankai University, Tianjin 300041, China; 3Department of Oral and Maxillofacial Surgery, Peking University School and Hospital of Stomatology, National Center of Stomatology and National Clinical Research Center for Oral Diseases, National Engineering Research Center of Oral Biomaterials and Digital Medical Devices, Beijing 100081, China

**Keywords:** cleidocranial dysplasia, delayed eruption of permanent teeth, lncRNA, osteoclastogenesis, bone resorption

## Abstract

Delayed eruption of permanent teeth is a common symptom of cleidocranial dysplasia (CCD). Previous studies have focused on the anomaly of osteogenesis resulting from mutations in the Runt-related transcription factor-2 gene (RUNX2). However, deficiencies in osteoclastogenesis and bone resorption, and the epigenetic regulation mediated by long non-coding (lnc)RNAs in CCD remain to be elucidated. Here, a novel osteoclast-specific lncRNA (OC-lncRNA) was identified during the osteoclast differentiation of RAW 264.7 cells transfected with a RUNX2 mutation expression cassette. We further confirmed that OC-lncRNA positively regulated osteoclastogenesis and bone resorption. The OC-lncRNA promoted the expression of CXC chemokine receptor type 3 (CXCR3) by competitively binding to microRNA (miR)-221-5p. The CXCR3–CXC-motif chemokine ligand 10 (CXCL10) interaction and nuclear factor-κB constituted a positive feedback that positively regulated osteoclastogenesis and bone resorption. These results demonstrate that OC-lncRNA-mediated osteoclast dysfunction via the OC-lncRNA–miR-221-5p–CXCR3 axis, which is involved in the process of delayed tooth eruption of CCD.

## 1. Introduction

Elucidating the etiology of disease processes not only enables novel therapeutics to be developed, but in some cases the abnormal may also illuminate the normal. Given the multiple disorders of hard tissue, patients affected with cleidocranial dysplasia (CCD) offer a valuable model for study of the osseous theme. Cleidocranial dysplasia is a rare congenital defect of skeleton and teeth which is inherited in an autosomal dominant manner. Dental anomalies are the main manifestations of CCD, with most patients showing the failure or delayed eruption of permanent teeth [1,2] which causes major physical and financial burdens. We previously reported ten independent cases of CCD who all presented with the delayed eruption of permanent teeth [3,4]. However, the molecular mechanism underlying the anomaly of tooth eruption in CCD has not been fully elucidated.

Alveolar bone resorption plays a pivotal role in tooth eruption [5], while other factors such as the “propulsive force” from tooth development appear to be of secondary importance. This is because once the eruption pathway has formed, even a metal crown lacking a developmental “propulsive force” can erupt on schedule [6]. The histological examination and imaging of alveolar bone has previously showed increased bone density in CCD, resulting in mechanical obstruction overlying the tooth crown [7,8,9]. Moreover, animal models of osteoclast dysfunction also presented with eruption defects [10,11,12]. It therefore appears that osteoclast and alveolar bone resorption are key to delayed tooth eruption in CCD [12,13].

Paradoxically, CCD has been considered a disorder characterized by deficient osteogenesis [14] because Runt-related transcription factor-2 (*RUNX2*) is the master gene involved in regulating osteoblast differentiation and osteogenesis [15], and various *RUNX2* mutations have been identified in about 70% of CCD patients [16]. It is conceivable that compromised bone resorption and osteoclast dysfunction have been overlooked as the cause of delayed or failed tooth eruption of CCD. Prior to the onset of tooth eruption, there is an influx of monocytes into the dental follicle, then the cells fuse to form osteoclasts [17]. It has not been elucidated whether *RUNX2* mutations directly impact the recruitment of monocytes and the subsequent formation and activation of osteoclasts through regulating signaling pathways. This is largely because most studies have focused on the reduced induction of osteoclast differentiation driven from osteoblasts and dental follicles [18,19], rather than the direct effect of *RUNX2* mutations within osteoclasts.

Recently, we showed that *RUNX2* directly regulates osteoclast differentiation and bone resorption through the classical nuclear factor of activated T-cells cytoplasmic 1 (NFATc1) signaling pathway [9] in the first report of the direct function of *RUNX2* in osteoclasts. In the present study, we focused on the molecular mechanism of the delayed eruption of permanent teeth in CCD, and related anomalies in bone remodeling. We identified a novel osteoclast-specific long non-coding (lnc)RNA, OC-lncRNA, by the sequencing of molecules involved in the deficient alveolar bone resorption of CCD. In this study, we examined the hypothesis that OC-lncRNA regulated osteoclastogenesis and bone resorption by the OC-lncRNA–microRNA (miR)-221-5p–CXC chemokine receptor type 3 (CXCR3) axis. The finding of the present study highlights the role for OC-lncRNA-mediated osteoclast dysfunction in the delayed tooth eruption of CCD.

## 2. Materials and Methods

### 2.1. Participants

Ten unrelated CCD patients were collected from Department of Preventive Dentistry, Peking University School of Stomatology. The diagnostic criteria refer to Mundlos’s report (Clinical features: brachycephaly; frontal and parietal bossing; open sutures and fontanelles; delayed closure of fontanelles; relative prognathism; soft skull in infancy; depressed nasal bridge; hypertelorism; ability to bring shoulders together; narrow, sloping shoulders; scoliosis; kyphosis; brachydactyly; tapering of fingers; nail dysplasia/hypoplasia; short, broad thumbs; clinodactyly of 5th finger; normal deciduous dentition; supernumerary teeth; impacted, supernumerary teeth; delayed eruption; crowding; malocclusion) [20]. Ethical approvals were obtained from the Ethical Committee of Peking University School and the Hospital of Stomatology (approval number: PKUSSIRB-2012004).

### 2.2. Cell Culture and Osteoclast Induction

Murine macrophage cell line RAW 264.7 was cultured in DMEM complete media (Gibco, Paisley, UK) containing 10% fetal bovine serum (FBS, Gibco, Carlsbad, CA, USA). Every 2 days, the medium was refreshed, and the cells were passaged. Differentiation medium, DMEM with 10% FBS and 10 ng/mL murine recombinant RANKL (R&D Systems, Minneapolis, MN USA), was used for osteoclast induction. Cells were cultivated in a humidified incubator at constant conditions with 37 °C, 5% CO_2_.

### 2.3. Total RNA Isolation and Real-Time PCR

Male C57BL/6 mice aged 6–8 weeks were sacrificed by cervical dislocation to obtain tissue samples (spleen, colon, skeletal muscle, bladder, kidney, lung, stomach, small intestine, pancreas, liver, brain, heart, and long bone). The total RNA from tissues and cells was extracted using TRIzol reagent (Thermo Fisher Scientific, Inc., Waltham, MA, USA) following the manufacturer’s protocol. The cDNA was generated from total RNA using the protocol of a reverse transcription kit (Thermo Scientific, Waltham, MA, USA). Amplification and detection of the specific products were performed using the SYBR Green PCR kit (Roche Applied Science, Indianapolis, IN, USA) on the ABI 7500 Real-time PCR System (Applied Biosystems, Carlsbad, CA, USA). As a housekeeping gene, GAPDH was used for the normalization.

### 2.4. RNA-seq and Bioinformatics Analysis

Cell lines WT-RAW and MUT-RAW were established as previously described [9]. The MUT-RAW cells were RAW 264.7 cells with c.514delT mutation (patient #5 in Table 1). After induction with RANKL for 3 days, total RNA was extracted from WT-RAW and MUT-RAW. The RNA quality control was performed using the Agilent bioanalyzer 2100 (Agilent Technologies, Santa Clara, CA, USA). The libraries construction and sequencing were performed by Berry Genomics (Beijing, China) with Illumina HiSeq 2000 sequencing platform. Genes with a fold change ≥ 2 and *p* < 0.05 were considered as the differentially expressed genes and selected for further network analysis. The logistic regression model CPAT that recognizes coding and noncoding transcripts [21], was used to predict the protein-coding potential.

### 2.5. FISH Assays

A Cy3-labeled probe mix specific targeted to OC-lncRNA was designed and synthesized by GenePharma Corporation (Shanghai, China). The hybridization was performed using a FISH kit (GenePharma) following the manufacturer’s instructions. Briefly, cells on coverslips were fixed with 4% paraformaldehyde for 15 min and washed with PBS. The fixed cells were incubated with hybridization buffer containing 20 μM FISH probe overnight at 37 °C. Cell nuclei were stained with DAPI and the confocal images were obtained with a confocal microscope (LMS710, Zeiss, Oberkochen, Germany).

### 2.6. Luciferase Assay

The Wt-OC-lncRNA, mut-OC-lncRNA, 3′-UTR of CXCR3 (wt), and 3′-UTR of CXCR3 (mut) were amplified using PCR and cloned into the pSI-check2 vector by Hanbio Biotechnology (Shanghai, China). The HEK-293T cells were seeded in 96-well plates and transfected using Lipofectamine 3000 regent (Invitrogen, Carlsbad, CA, USA). After 48 h post-transfection, the relative luciferase activity was measured using a dual-luciferase reporter assay system (Promega Corp., Madison, WI, USA).

### 2.7. RNA Pull-Down Assay

A RNA pull-down assay was performed as described in Subramanian et al. [22]. Briefly, biotinylated NC mimic or miR-221-5p mimic was transfected in RAW 264.7 cells. After 48 h, the cell lysates were incubated with streptavidin magnetic beads at 4 °C for 4 h. The RNA isolation was performed using the acid-phenol: chloroform. The OC-lncRNA present in the pull-down material was detected by qRT-PCR analysis.

### 2.8. Western Blot Analyses

Total protein was extracted using RIPA buffer (Huaxing Bio, Beijing, China) containing protease inhibitors. Cytoplasmic and nuclear protein extracts were prepared using the Nuclear and Cytoplasmic Extraction Reagents (Thermo Scientific) following the manufacturer’s instructions. Equal amounts of protein samples were loaded equally in 10% SDS-PAGE gels and then transferred to PVDF membranes (Millipore, Billerica, MA, USA). After being blocked with 5% skim milk for 1 h at room temperature, the membranes were incubated with primary antibodies at 4 °C overnight followed by incubation with secondary antibody coupled to HRP for 1 h (1:10,000 dilution). The following primary antibodies were used at a dilution of 1:1000: GAPHD (Huaxing Bio, Beijing, China); NF-κB p65 (CST, MA, USA); MMP9, CXCR3 and Lamin B1 (Proteintech, Wuhan, China); CTSK and CD14 (Abcam, Cambridge, UK). The Bio-Rad Quantity One software was used to quantify the densitometry of bands.

### 2.9. TRAP Staining

The RAW 264.7 cells (1.5 × 10^4^ cells/mL) were seeded onto 24-well plates. At 4 days after osteoclast induction, the cells were fixed in 4% paraformaldehyde and stained using a TRAP staining kit (Sigma–Aldrich, St. Louis, MO, USA) according to the manufacturer’s protocol. A positively stained cell with three or more nuclei was defined as an osteoclast. The number of osteoclasts was counted under a light microscope manually.

### 2.10. Bone Resorption Assay

The RAW 264.7 cells were plated onto Corning Osteo assay Surface in a 24-well plate (Corning, Corning, NY, USA) at a density of 1.5 × 10^4^ per well. After incubation with 10 ng/mL RANKL for 7 days, the cells were removed using 10% sodium. The resorption pits were imaged under a light microscope and analyzed using Image-Pro Plus (version 6.2.0.424, Media Cybernetics, Inc., Silver Spring, MD, USA).

### 2.11. Immunohistochemistry

Alveolar bone samples were collected from two CCD patients and three healthy controls in the guided eruption surgery. Alveolar bone samples were fixed in 10% buffered formalin and decalcified in 10% buffered EDTA (pH 7.4). Then the samples were embedded in paraffin and sectioned at 5 μm thickness.

Sections were incubated overnight with primary antibody to CXCR3 (Proteintech, Wuhan, China, 1:200 dilution) and CD14 (Abcam, Cambridge, UK, 1:200 dilution) at 4 °C. After being washed, the sections were stained with HRP-conjugated secondary antibody for 20 min at room temperature. Images were obtained using a light microscopy (BX51, Olympus, Shinjuku City, Japan). Quantification of protein intensity was performed using Image-Pro Plus 6.0 software (Media Cybernetics, Silver Spring, MD, USA) with integral optical density (IOD).

### 2.12. ELISA

Cell culture supernatants were collected at indicated time points (2 h, 4 h, 8 h, 12 h, 24 h, 48 h, and 72 h). The amount of CXCL10 in the supernatants was determined using an ELISA kit (Abcam, Cambridge, UK) according to the supplier’s protocol.

### 2.13. Transient Transfection

The RNA oligonucleotides (miR mimics, miR inhibitors, and siRNAs) were designed and synthesized by RiboBio Company (Guangzhou, China). The pCDNA3.1-CXCR3 plasmid was constructed and synthesized by Tsingke Biological Technology (Beijing, China). Lipofectamine 3000 was used for transfection following the manufacturer’s instructions. Further analysis or treatments were processed at least 48 h after transfection.

### 2.14. Transwell Assay

Transwell assays were performed to investigate the migration of monocytes. Monocytes were harvested and suspended in serum-free media (10^5^ cells/mL). Then cell suspension was plated into the upper chamber of a 24-well transwell chambers (pore size 8 μm; Corning, Corning, NY, USA), 100 μL per chamber. The lower chamber contained 500 μL complete growth media with 100 ng/μL CXCL10. After incubation for 12 h, the upper surface of the membrane was carefully wiped with a cotton swab, the migrated cells on the bottom side of the membrane were fixed with 4% paraformaldehyde and stained with 0.1% crystal violet (Beyotime Biotechnology, Jiangsu, China). The stained cells were photographed and counted using a bright field microscope (BX51, Olympus, Japan).

### 2.15. RACE Assay

The RACE assay was performed to determine the 5′- and 3′-end sequences of TCONS_00045498. The SMARTer^®^ RACE 5′/3′ Kit (TAKARA, Tokyo, Japan) was used according to the manufacturer’s instructions. The RACE PCR products were separated on a 1.5% agarose gel. The amplified bands were sequenced bi-directionally. The gene-specific primers for RACE analysis are listed in Appendix A.

### 2.16. LncRNA Knock-Out by CRISPR/Cas9 Technique

The CRISPR/Cas9 technique was used for the generation of OC-lncRNA knock-out RAW 264.7 cell line. Two-round lentivirus infections were performed for the exogenous expression of the specific sgRNA and Cas9 protein in RAW 264.7 cells. Several single clones were then selected and expanded for follow-up experiments. The PCR and direct sequencing confirmed the successful deletion of OC-lncRNA. The expression levels of OC-lncRNA and the relative genes after OC-lncRNA knock-out were analyzed by real-time PCR. The sequences of gRNAs are listed in Appendix A.

### 2.17. Statistical Analysis

Results are presented as mean and standard deviation. Student’s *t*-test and one-way analysis of variance analyses were performed when appropriate. All significance levels were set at *p* < 0.05.

## 3. Results

### 3.1. Delayed Eruption of Permanent Teeth in CCD Primarily Results from Deficient Alveolar Bone Resorption

The 10 independent CCD patients we collected all presented with typical phenotypes including the delayed eruption of permanent teeth (Table 1). Compromised tooth eruption in CCD predominantly results from decreased alveolar bone resorption leading to incomplete formation of the eruption pathway. After eliminating the coronal obstruction of dense alveolar bone, the impacted teeth erupted spontaneously without orthodontic traction. In CCD patient #5, the impacted first premolars erupted actively after removal of the alveolar bone overlying the dental follicle by surgical exposure (Figure 1A,B). The roots of the impacted teeth were shown to form normally (Figure 1C), confirming that tooth impaction in CCD was caused by deficient bone resorption. Additionally, tartrate-resistant acid phosphatase (TRAP) staining showed that *Runx2* knock-out in RAW 264.7 cells greatly impaired osteoclast differentiation (Appendix A).

### 3.2. A Novel Osteoclast-Specific lncRNA Positively Regulates Osteoclast Differentiation and Bone Resorption

To investigate global changes in gene expression arising from *RUNX2* mutations, we performed RNA sequencing of osteoclasts induced by wild-type *RUNX2* transfected RAW 264.7 cell line (WT-RAW) and mutant *RUNX2* transfected RAW 264.7 cell line (MUT-RAW). Bioinformatics analysis revealed a regulatory network of differentially expressed mRNAs, lncRNAs, miRNAs, and circular RNAs (Figure 2A). Ten lncRNAs showed a broad role in the regulation of multiple mRNAs (Figure 2B). Among these, lncRNA TCONS_00045498 was selected for further investigation based on combined bioinformatics predictions and expression validation by real-time PCR (Figure 2C). The 5′- and 3′-rapid amplification of complementary cDNA Ends (RACE) analyses identified a 976-bp full-length transcript of TCONS_00045498 (Figure 2D). A BLAST search of the National Center for Biotechnology Information (NCBI) database showed this transcript to be located on mouse chromosome 2 (116,897,765–116,896,790). This novel transcript variant of lncRNA Gm46758 contains four exons (Figure 2E) and shares a 268-bp sequence with the other three transcripts (X5, X2, and X1) included in the NCBI database (Figure 2F). We named the novel transcript OC-lncRNA (Osteoclastic lncRNA) and its sequence is indicated in Figure 2F. Coding-Potential Assessment Tool (CPAT) analysis found that OC-lncRNA had no protein-coding potential and should be defined as a non-coding RNA (Figure 2G,H).

Following NF-κB ligand (RANKL)-induced osteoclast differentiation, OC-lncRNA expression significantly increased compared with day 0 and peaked on day 3, then decreased gradually (Figure 3A). Real-time PCR evaluation of OC-lncRNA expression in various C57BL/6 mouse tissues and four common mouse cell lines (M3T3-E1, NIH3T3, C2C12, and RAW 264.7) revealed markedly high expression in long bone compared with other tissues. Significantly higher OC-lncRNA expression was also detected in the murine macrophage cell line RAW 264.7 compared with other common murine cell lines, including the osteoblastic cell line MC3T3-E1 (Figure 3B). These results suggest that OC-lncRNA is a bone tissue-specific lncRNA, and likely to be osteoclast-specific. Because the cellular localization of lncRNAs typically reflects their biological activities, we used RNA fluorescence in situ hybridization (FISH) of OC-lncRNA to determine its subcellular location in RAW 264.7 cells. Under non-induced conditions, OC-lncRNA was mainly localized in the RAW 264.7 cell cytoplasm (Figure 3C). It remained in the cytoplasm of osteoclasts during RANKL-induced osteoclasts differentiation and formation, with no obvious nuclear translocation (Figure 3D). These results suggest that OC-lncRNA participates in signaling regulation in the cytoplasm.

To further investigate its regulatory function, OC-lncRNA was knocked-out in RAW 264.7 cells by clustered regularly interspaced short palindromic repeats (CRISPR)/CRISPR-associated protein 9 (Cas9) technique. The PCR amplification and sequence alignment showed that OC-lncRNA was successfully knocked out in one clone (data not shown) with an efficiency above 95% as verified by real-time PCR compared with controls (Appendix A). After RANKL induction, KO-OC-lncRNA RAW 264.7 cells showed a dramatic reduction in the number and size of the TRAP positive osteoclasts compared with control cells (Figure 3E). Consistent with TRAP staining, a severe reduction in the bone-resorbing function of KO-OC-lncRNA osteoclasts was observed, with a significant decrease in the total bone resorption area in KO-OC-lncRNA osteoclasts compared with controls (Figure 3F). Real-time PCR and western blotting showed that genes associated with bone resorption (cathepsin K [CTSK] and matrix metallopeptidase [MMP]9) were downregulated at both mRNA and protein levels following OC-lncRNA knock-out (Figure 3G,H). These results demonstrate that OC-lncRNA positively regulates osteoclast differentiation and bone resorption.

### 3.3. OC-lncRNA Positively Regulates Osteoclasts via a Competing Endogenous (ce)RNA Mechanism

Because OC-lncRNA is localized to the cytoplasm, ceRNA was the primary consideration for its regulatory mechanism. Bioinformatics analysis showed that OC-lncRNA and CXCR3 formed a ceRNA module that bound competitively to miR-221-5p (Appendix A). In contrast to the observed OC-lncRNA expression trend, miR-221-5p expression was lowest on day 3 and slightly increased on day 5 in RANKL-induced osteoclast differentiation (Appendix A). Binding sites of OC-lncRNA and miR-221-5p were predicted by complementary base pairing (Appendix A). The dual-luciferase reporter assay found that the overexpression of miR-221-5p and wt-OC-lncRNA, but not that of miR-221-5p and mut-OC-lncRNA, reduced luciferase activity. By contrast, normal control (NC) mimic overexpression had no impact on the luciferase activity of wt-OC-lncRNA or mut-OC-lncRNA (Figure 4A). Subsequently, RNA pull-down assay was performed to further demonstrate the binding between OC-lncRNA and miR-221-5p. Biotinylated NC mimic (bio-miR-NC) or miR-221-5p mimic (bio-miR-221) was transfected in RAW 264.7 cells and then pulled down by streptavidin magnetic beads, as well as the OC-lncRNA enriched. The OC-lncRNA enrichment was detected by qRT-PCR analysis. We found that the OC-lncRNA enrichment of biotinylated miR-221-5p was higher than that of biotinylated NC mimic (Figure 4B). Additionally, miR-221-5p expression was significantly higher in KO-OC-lncRNA RAW 264.7 cells than that in controls (Figure 4C). These results suggest that OC-lncRNA acts as a miR-221-5p sponge.

Subsequently, CXCR3 expression levels during osteoclast differentiation were detected by real-time PCR and western blotting. On day 3, CXCR3 expression was significantly increased at both the mRNA and protein level compared with day 0 and followed by a gradual decrease on day 5 (Appendix A). This expression pattern was the opposite of miR-221-5p during osteoclast differentiation. Binding sites of miR-221-5p and the 3′- untranslated region (UTR) of CXCR3 were predicted by complementary base pairing (Appendix A). The dual-luciferase reporter assay indicated that miR-221-5p binding to the wt-3′-UTR of CXCR3 decreased luciferase activity, while the co-transfection of miR-221-5p and the mut-3′-UTR of CXCR3 had no effect on activity (Figure 4D). In support of this, overexpressed miR-221-5p mimics transiently transfected into RAW 264.7 cells reduced CXCR3 protein expression (Figure 4E), while miR-221-5p inhibitor overexpression upregulated CXCR3 protein expression (Figure 4F). These findings imply that CXCR3 is a target gene of miR-221-5p, and that OC-lncRNA and the 3′-UTR of CXCR3 competitively bind to miR-221-5p in a ceRNA model.

### 3.4. CXCR3 Is a Positive Regulatory Gene in Osteoclast Differentiation and Bone Resorption

The CXCR3 is a chemokine receptor that plays a pivotal role in immunity and inflammation. However, its functions in osteoclast differentiation and bone resorption have not been fully elucidated. We knocked down CXCR3 in RAW 264.7 cells by small interfering (si)RNA using three different targeting siRNAs. The siRNA #3 showed the best interference effect (Figure 5A,B) and was used for subsequent experiments. The CXCR3 knockdown group showed a clear reduction of TRAP+ osteoclasts (Figure 5C) and significantly fewer osteoclasts compared with the control (Figure 5D). Additionally, the CXCR3 knock-down group showed reduced mRNA and protein expression of the key proteolytic enzymes CTSK and MMP9 (Figure 5E,F), as well as a reduced area of resorption pits compared with the control (Figure 5G,H). These results indicate that CXCR3 plays a positive regulatory role in osteoclast differentiation and bone resorption.

### 3.5. OC-lncRNA Reduction Diminishes Monocyte Chemotaxis through CXCR3 Downregulation

Because of the observed competitive binding of OC-lncRNA and CXCR3 to miR-221-5p, we speculated that CXCR3 downregulation would impair the chemotaxis of monocytes towards the CXC-motif chemokine ligand 10 (CXCL10). We evaluated the impact of OC-lncRNA reduction on monocyte chemotaxis using a transwell assay, which detected a decrease in the number of KO-OC-lncRNA RAW 264.7 cells migrating to the lower chamber containing CXCL10 compared with normal control cells (Figure 6A,B). This prompted us to consider that the migration of monocytes to the alveolar bone in remodeling may be obstructed in CCD patients because of the reduction of OC-lncRNA. Immunohistochemical staining of CXCR3 and CD14 was used to indirectly assess the number of monocytes migrating to alveolar bone overlying the dental follicle. In the alveolar bone of healthy controls, staining for CXCR3 and CD14 was more intense than in CCD patients (Figure 6C,E). The quantification of protein expression also showed that CXCR3 and CD14 were more highly expressed in the alveolar bone of healthy controls compared with CCD patients (Figure 6D,F). These results suggest that OC-lncRNA reduction impairs monocyte chemotaxis via CXCR3 downregulation.

### 3.6. OC-lncRNA Facilitates Osteoclast Differentiation through CXCL10–CXCR3 Autocrine Signaling

The NF-κB expression and activation are important for RANKL-mediated osteoclastogenesis [23], while CXCL10 increases expression of the NF-κB subunit p65 and promotes NF-κB activity [24]. We used the enzyme-linked immunosorbent assay (ELISA) to detect CXCL10 secreted in cellular supernatants to investigate the precise molecular mechanism of CXCL10–CXCR3 in osteoclast differentiation and bone resorption. Secretion of CXCL10 was gradually increased during the RANKL-induced osteoclast differentiation of RAW 264.7 cells. At 72 h, CXCL10 levels in the supernatant were greatly elevated compared with 2 h (Figure 7A).

The CXCL10 may be involved in osteoclast differentiation by binding to its receptor CXCR3 which is anchored to the cell membrane. Considering that CXCR3 is downregulated in KO-OC-lncRNA RAW 264.7 cells, we next examined the secretion of CXCL10 from these cells induced with RANKL. As expected, throughout the early stages of differentiation, CXCL10 secretion by KO-OC-lncRNA RAW 264.7 cells was lower compared with the control (Figure 7B). This indicates that signaling pathways mediated by CXCL10–CXCR3 could be involved in the dysfunction of KO-OC-lncRNA RAW 264.7 cells.

In normal RAW 264.7 cells, western blotting revealed that CXCL10 stimulation for 30 min induced the nuclear translocation of p65 (Figure 7C). However, p65 translocation was reduced in KO-OC-lncRNA RAW 264.7 cells compared with the controls (Figure 7D). Reduced CXCR3 could prevent the signaling activation of CXCL10–CXCR3, causing osteoclastogenesis malfunction in KO-OC-lncRNA RAW 264.7 cells, so we investigated this by overexpressing CXCR3 in KO-OC-lncRNA RAW 264.7 cells using transient transfection (Figure 7E). After RANKL induction, CXCR3 overexpression promoted osteoclast differentiation to some extent compared with the control group transfected with an NC plasmid (Figure 7F,G). Similarly, the bone resorptive capacity was elevated in KO-OC-lncRNA RAW 264.7 cells after CXCR3 overexpression (Figure 7H,I). These results suggest that CXCL10–CXCR3 autocrine signaling regulates osteoclast differentiation and bone resorption by promoting the nuclear translocation of p65. As a ceRNA of CXCR3, OC-lncRNA plays a protective role in CXCR3 expression. A proposed model of the signaling pathway and biological process mediated by OC-lncRNA in tooth eruption is illustrated in Figure 7J.

## 4. Discussion

The present study investigated osteoclast dysfunction and dysdifferentiation in patients affected with CCD. We also identified a novel osteoclast-specific lncRNA that regulates early-stage osteoclastogenesis and bone resorption via the OC-lncRNA–miR-221-5p–CXCR3 axis, which confirmed our initial hypothesis. The OC-lncRNA serves as a miR-221-5p sponge to ensure CXCR3 expression, which plays a crucial role in the chemotaxis and differentiation of osteoclast precursors. The CXCL10–CXCR3 interaction promotes bone resorption in tooth eruption in two ways: stimulating the migration of osteoclast precursors to eruptive sites and facilitating the formation and maturation of osteoclasts through the CXCL10–CXCR3–NF-κB positive feedback loop.

Tooth eruption is a complex and tightly regulated biological process [5] that requires diverse cellular coordination with strict temporal and spatial patterns [25]. Formation of the eruption pathway is of fundamental importance in tooth eruption because it mechanically determines whether a tooth can emerge in the oral cavity. It mainly relies on the resorption of alveolar bone overlying the dental follicle, rather than the movement of teeth or osteogenesis [26]. Almost all CCD patients, including the 10 we collected, suffer from a marked delay or failure in eruption because of the dense alveolar bone overlying the dental follicle resulting from deficient bone resorption [9]. Because the impacted tooth is obstructed by bone resistance, an understanding of the biology of bone resorption in the formation of an eruption pathway is crucial [5]. Our clinical findings of CCD patients showed active eruption of the impacted teeth accompanied by root formation after the removal of bone resistance, without orthodontic traction. Therefore, CCD offers us a valuable model to study molecular relationships and signaling pathways involved in the formation, maturation, and activation of osteoclasts, as well as bone resorption.

The findings of the present study extend our comprehension of the biological process of tooth eruption and the regulatory genes involved, as well as our knowledge about CCD. Cleidocranial dysplasia is more than a purely osteogenic anomaly, but also involves the aberrant regulation and dysfunction of osteoclasts. They may also pave the way for the treatment of osteoclast-related diseases, such as postmenopausal osteoporosis and bone metastasis of specific cancers which arise from excessive bone resorption, and disorders arising from deficient bone resorption such as osteosclerosis and pycnodysostosis [27].

This study is also the first to demonstrate that epigenetic regulation mediated by lncRNA is involved in the pathogenesis and molecular mechanism of CCD. The LncRNAs are defined as non-protein-coding transcripts longer than 200 nucleotides in length [28]. They perform a range of diverse functions, and are extensively implicated in cell differentiation and organ development [29]. This diversity and organismal complexity have posed a considerable challenge for studying their functional roles and mechanisms of action.

Liu et al. [30] and Dou et al. [31] respectively described lncRNA expression profiles during osteoclastogenesis of human and mouse osteoclasts. In both human and mouse cells, hundreds of lncRNAs were shown to be significantly altered in osteoclast differentiation. Gene ontology and Kyoto Encyclopedia of Genes and Genomes pathway analyses and the lncRNA–mRNA co-expression network indicated that lncRNAs extensively modulate gene expression and signal transduction during different stages in osteoclastogenesis. In the present study, we identified a novel lncRNA that is highly expressed in bone tissue, especially in osteoclasts, but not osteoblasts. Mechanistically, the OC-lncRNA–miR-221-5p–CXCR3 axis modulates osteoclastogenesis and the bone-resorbing function of osteoclasts. The OC-lncRNA knock-out clearly obstructed the formation and maturation of osteoclasts, as well as bone resorption, demonstrating that OC-lncRNA plays a critical role in osteoclastogenesis. These findings are consistent with the initial hypothesis that, OC-lncRNA regulated osteoclastogenesis and bone resorption. Moreover, OC-lncRNA localized in the cytoplasm of osteoclasts and fulfilled a regulatory role via the classic ceRNA model. During early stages of osteoclast differentiation, increased levels of OC-lncRNA offered a protective effect on the sufficient expression of CXCR3 through sponging miR-221-5p.

Sequence competition is prevalent in gene regulation [32]. The LncRNA MALAT1, deriving from exosomes of endothelial progenitor cells, was reported to enhance the recruitment and differentiation of osteoclast precursors via the MALAT1–miR-124–integrin subunit β1 axis [33]. Similarly, lncRNA MIRG also positively regulates osteoclastogenesis by the MIRG–miR-1897–NFATc1 axis [34]. Abnormal ceRNA regulation may contribute to the initiation and development of disease [32]. However, current evidence is still too limited to comprehensively determine the molecular mechanism of lncRNAs involved in the differentiation and maturation of osteoclasts. Nevertheless, individual research in this area may provide information for future investigation.

The receptor CXCR3 is a G protein-coupled, interferon-inducible chemokine receptor with three ligands: CXCL9, CXCL10, and CXCL11 [35]. Substantial evidence indicates that CXCR3 binding to its ligands has a modulatory role in immunity, inflammation, and oncogenesis [36,37,38]. The CXCL10–CXCR3 exerts multiple biological functions through paracrine and/or autocrine signaling. In the pathological bone destruction of rheumatoid arthritis, the reciprocal amplification of CXCL10 and RANKL was reported to promote osteoclast differentiation [39]. Additionally, CXCR3 gene silencing inhibited bone metastasis primarily through CXCL10 expression which occurs at much higher levels in bone than CXCL9 and CXCL11 [40]. Bone resorption in tooth eruption is a biological and genetically controlled progress not involving inflammation or infection [25]. However, the exact interplay and roles of CXCL10–CXCR3 in the recruitment and differentiation of monocytes during bone resorption remain elusive. Moreover, to our knowledge, there is no report about the regulatory role of CXCL10–CXCR3 in the physiological bone remolding of tooth eruption.

The present study is the first to demonstrate that CXCR3 positively regulates osteoclastogenesis and bone resorption in tooth eruption. In the early stage of osteoclast differentiation, CXCR3 and its ligand CXCL10 were significantly increased and exerted functions in an autocrine manner. Additionally, exogenous CXCR3 overexpression partially ameliorated the formation and activation of osteoclasts in OC-lncRNA knock-out monocytes. We showed that CXCL10 and CXCR3 binding promoted nuclear translocation of the NF-κB p65 subunit, which is consistent with the findings of Jin et al. in breast cancer cells [24]. The NF-κB is a crucial transcription factor in the early stage of osteoclastogenesis [41] whose nuclear translocation is a prerequisite for transcriptional activation [42]. The NF-κB nuclear accumulation induced the expression of a series of key genes in signaling cascades, promoting the formation and activation of osteoclasts. In turn, NF-κB also increased CXCL10 expression, which formed a positive feedback loop [24,43], in which NF-κB directly binds to the promoter of CXCL10 and activates the transcription [44,45]. Similarly, CXCL10 expression was downregulated by suppressing NF-κB signaling pathways [46]. In the present study, we detected continually increasing levels of secreted CXCL10 in culture supernatants and showed that the nuclear translocation of p65 was promoted by CXCL10–CXCR3 binding.

Besides promoting osteoclast differentiation via CXCL10–CXCR3–NF-κB loop, the transwell migration assay and immunohistochemical staining showed that osteoclast-secreted CXCL10 also recruited CXCR3+ monocytes to bone-resorbing sites. More CXCR3+ CD14+ monocytes were observed in the alveolar bone of healthy controls, while chemotaxis of monocytes was attenuated in CCD patients because of the downregulation of CXCR3 and CXCL10 resulting from OC-lncRNA reduction. Similarly, knock-out or siRNA interference of CXCR3 also blocked the CXCL10-induced migration of bone marrow-derived macrophages [47]. Monocyte differentiation and recruitment deficiencies eventually reduce the number of osteoclasts in alveolar bone overlying the dental follicle and increased the bone density, as stated previously [9], followed by delayed tooth eruption (Figure 7J). The exogenous expression of CXCR3 in OC-lncRNA knock-out osteoclast precursors only partially rescued the differentiation and resorbing function of osteoclasts, likely because other crucial signaling pathways in osteoclastogenesis and bone resorption were also affected by OC-lncRNA knock-out. Further study is required to fully elucidate the molecular mechanism of OC-lncRNA in CCD, especially regarding epigenetic regulation. Further investigations are required to develop reliable analytical methods for demonstrating the function of OC-lncRNA in vivo. A transgenic mouse model with OC-lncRNA knockout is helpful in proceeding with in vivo experiments. Meanwhile, more CCD patients should be collected to further verify our findings.

In conclusion, the present study identified a novel osteoclast-specific lncRNA that positively regulates osteoclastogenesis and bone resorption via the OC-lncRNA–miR-221-5p–CXCR3 axis. The dysregulated formation and maturation of osteoclasts mediated by OC-lncRNA contributes to the delayed tooth eruption seen in CCD. Future novel insights into the molecular mechanism underlying OC-lncRNA could aid the development of new therapeutic targets for osteoclast-related diseases.

## 5. Conclusions

The novel OC-lncRNA was involved in the dysfunction of osteoclasts in CCD. The OC-lncRNA increased the chemotaxis and differentiation of osteoclast precursors and formation of the tooth eruption pathway; OC-lncRNA also promoted the CXCR3-CXCL10 interaction and the up-regulation of NF-κB through competitive binding to microRNA-221-5p. These findings will not only provide us with a better understanding of the etiology of tooth impaction in CCD but will also illuminate the precise molecular mechanism of osteoclastogenesis and bone resorption, aiding the development of effective CCD therapy.

## Figures and Tables

**Figure 1 cells-11-02729-f001:**
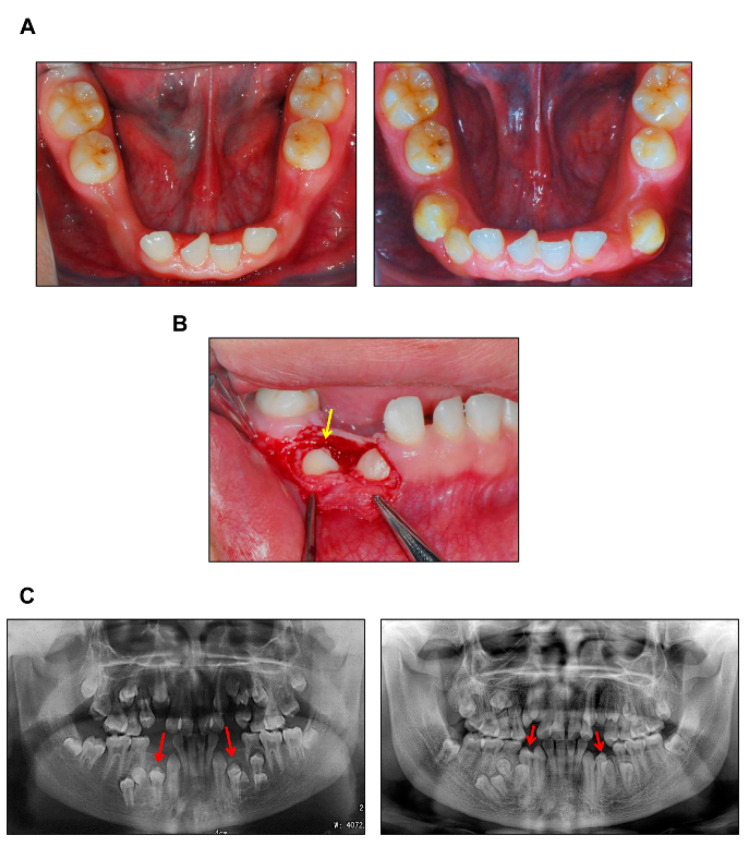
Guided eruption surgery of CCD patient. (**A**) Intraoral photographs of CCD patient #5. Left, pre-treatment, 12-year-old; Right, post-treatment, 19-year-old. (**B**) Surgical exposure of the impacted teeth, 16-year-old. Arrow: right mandibular first premolar. (**C**) Panoramic radiographs of CCD patient #5. Left, pre-treatment, 12-year-old; Right, post-treatment, 19-year-old. Arrows: mandibular first premolars.

**Figure 2 cells-11-02729-f002:**
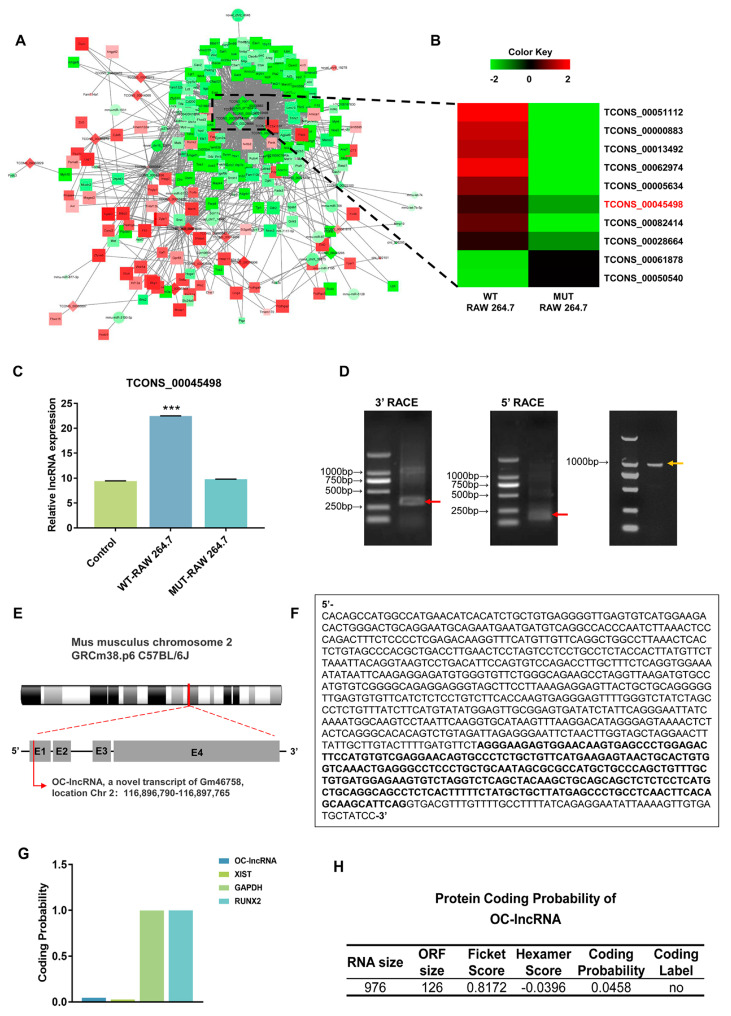
Identification of a novel lncRNA transcript in CCD. (**A**) The regulatory network of differentially expressed mRNAs (rectangles), lncRNAs (rhombuses), microRNAs (circles), and circRNAs (triangles). (**B**) The heatmap of 10 core regulatory lncRNAs. (**C**) Expression validation of TCONS_00045498 by real-time PCR with RANKL induction for 3 days. (**D**) The gel results of the 3′/5′ RACE products (red arrows) and the whole sequence (yellow arrow). (**E**) Schematic of OC-lncRNA. (**F**) The complete nucleic acid sequence of OC-lncRNA. Identical sequence among OC-lncRNA and three transcripts of Gm46758 is indicated in bold. (**G**) Coding-potential assessment of OC-lncRNA by CPAT analysis. (**H**) OC-lncRNA displayed no coding probability. LncRNA XIST was used as a representative non-coding gene. GAPDH and RUNX2 were used as representative coding genes. Error bars represent SD. *** *p* < 0.001.

**Figure 3 cells-11-02729-f003:**
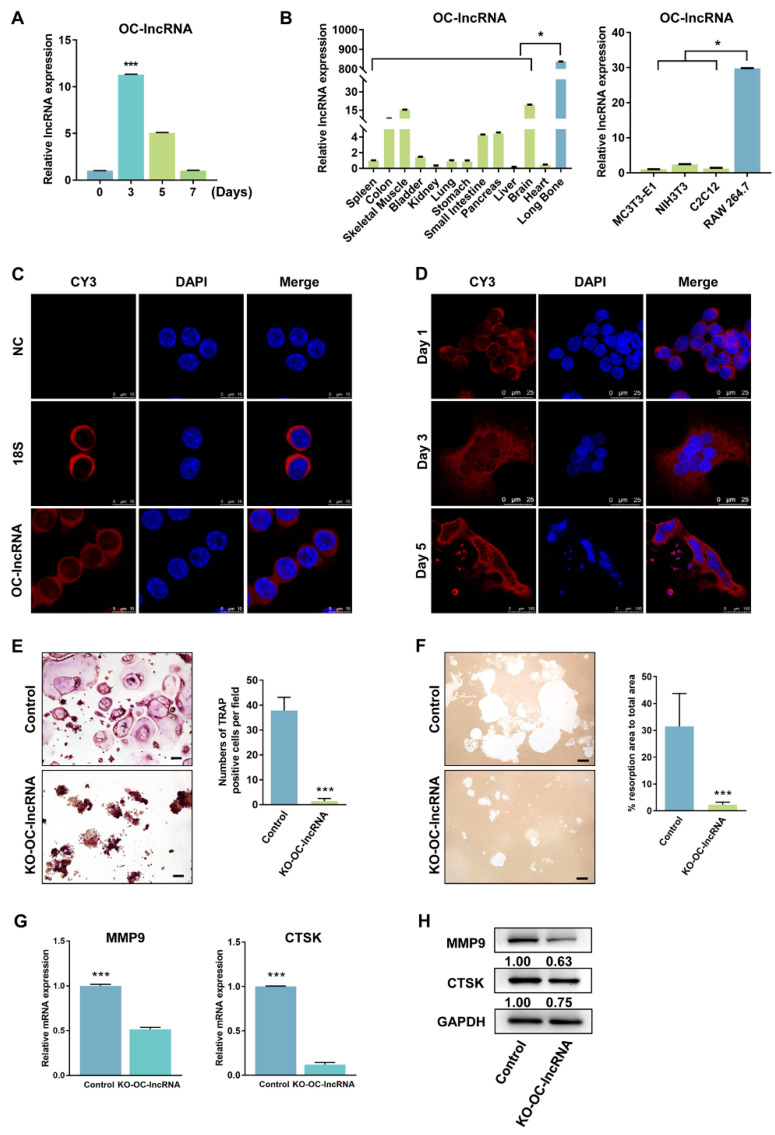
OC-lncRNA is an osteoclast-specific lncRNA localizing to the cytoplasm and positively regulates osteoclast differentiation. (**A**) Expression levels of OC-lncRNA in osteoclast differentiation and maturation at indicated time points. (**B**) Relative OC-lncRNA expression levels in various tissues of C57BL/6 mouse and four common mouse cell lines. (**C**) RNA-FISH for OC-lncRNA in non-treated RAW 264.7 cells. (**D**) RNA-FISH for OC-lncRNA at indicated time points of osteoclast differentiation. (**E**) TRAP staining of OC-lncRNA knock-out osteoclasts and control. Quantification of TRAP+ osteoclasts. (**F**) Representative images of bone resorption assay and quantification of resorption pits. (**G**) mRNA levels of osteoclastic genes levels detected by real-time PCR. (**H**) Osteoclastic protein levels detected by western blotting. Error bars represent SD. * *p* < 0.05, *** *p* < 0.001.

**Figure 4 cells-11-02729-f004:**
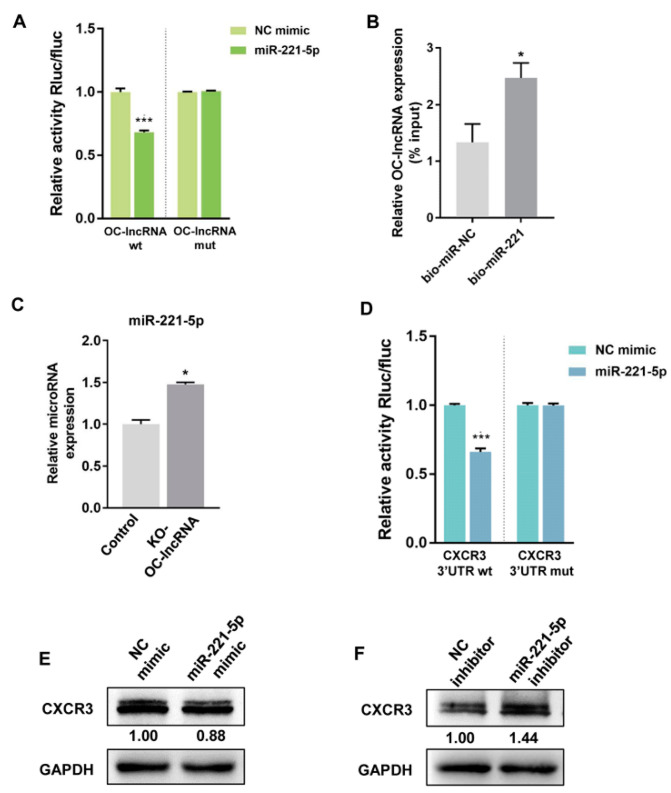
miR-221-5p bound to OC-lncRNA and CXCR3 mRNA. (**A**) Wt-OC-lncRNA or mut-OC-lncRNA were co-transfected with miR-221-5p mimics or NC mimics in HEK-293T cells. Luciferase report activity was detected. (**B**) OC-lncRNA was pulled down by biotin-labeled NC mimic (bio-miR-NC) or miR-221-5p (bio-miR-221) to determine the interaction between OC-lncRNA and miR-221-5p. (**C**) miR-221-5p expression level in KO-OC-lncRNA RAW 264.7 cells. (**D**) Wt-3′-UTR or mut-3′-UTR of CXCR3 were co-transfected with miR-221-5p mimics or NC mimics in HEK-293T cells. Luciferase report activity was detected. Protein levels of CXCR3 after overexpression miR-221-5p mimics (**E**) or inhibitor (**F**). Error bars represent SD. * *p* < 0.05, *** *p* < 0.001.

**Figure 5 cells-11-02729-f005:**
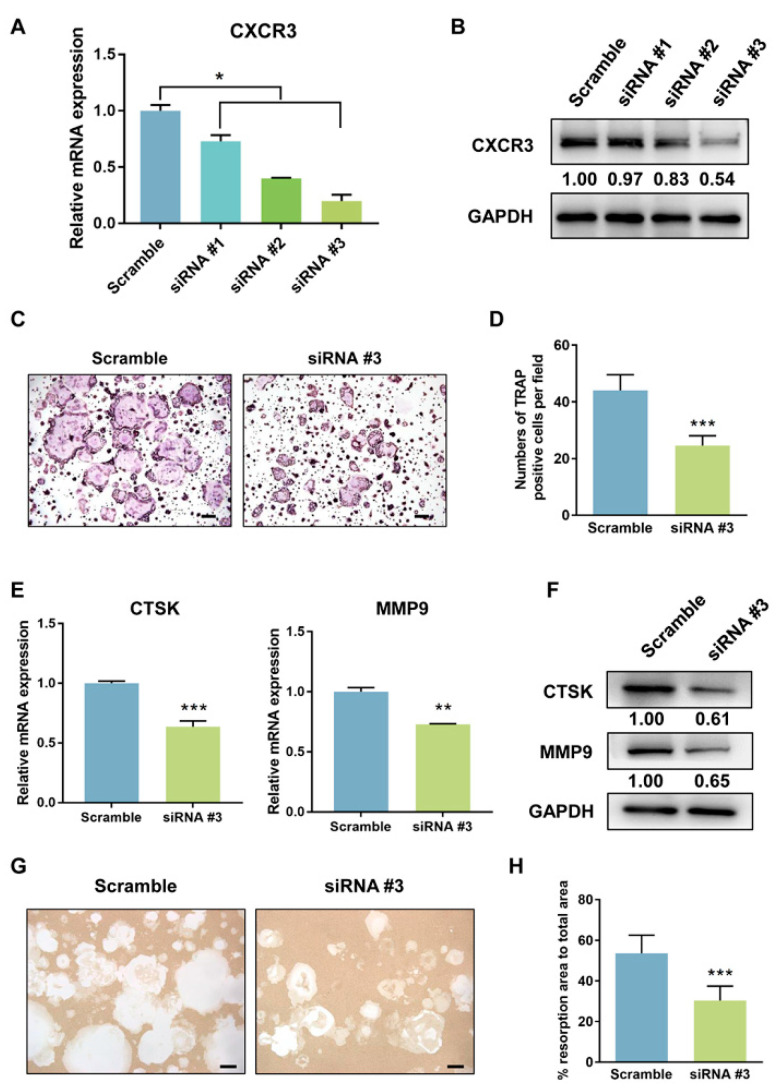
CXCR3 positively regulates osteoclast differentiation. The interference efficiency of the siRNAs targeting CXCR3 was detected on mRNA levels by real-time PCR (**A**) and protein levels by western blotting (**B**). (**C**) TRAP staining identified the osteoclast formation after CXCR3 knockdown. (**D**) Numbers of TRAP+ osteoclasts. Expression levels of genes related to bone resorption (CTSK and MMP9) were detected by real-time PCR (**E**) and western blotting (**F**). (**G**) Representative images of bone resorption assay after CXCR3 knockdown. (**H**) Quantification of resorption pits. Error bars represent SD. * *p* < 0.05, ** *p* < 0.01, and *** *p* < 0.001.

**Figure 6 cells-11-02729-f006:**
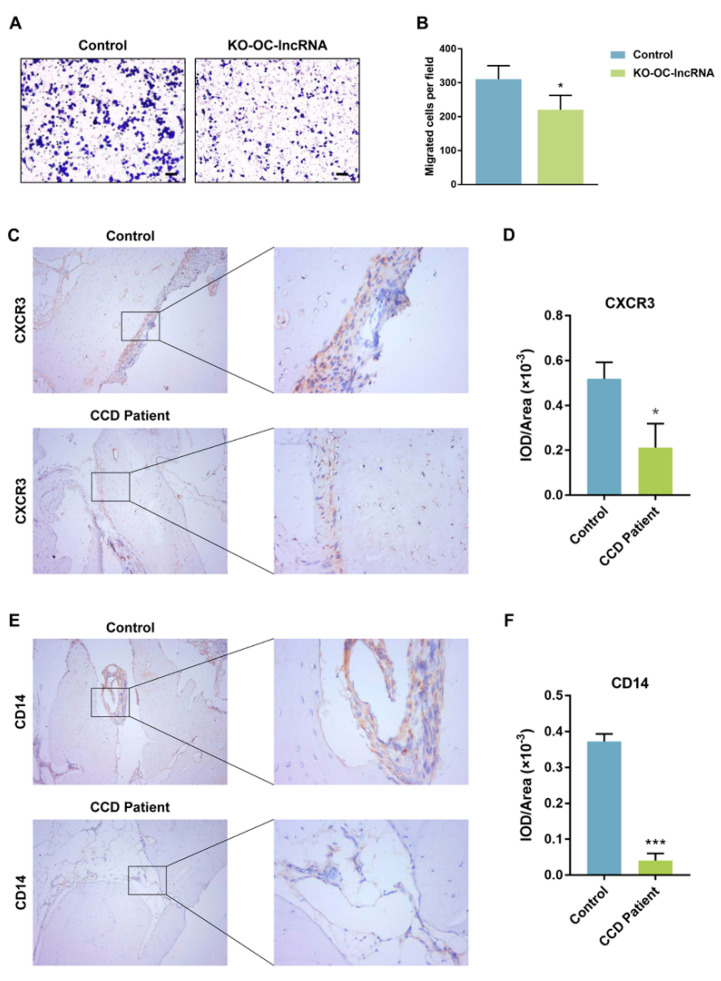
Monocytes chemotaxis detected by and immunohistochemistry. (**A**) Representative images of transwell migration assay. Scale bar, 500 μm. (**B**) Quantitative analysis of transwell migration. (**C**) Immunohistochemistry for CXCR3 in alveolar bone (Left panel, ×100; Right panel, ×400). (**D**) Quantification of CXCR3 protein expression. (**E**) Immunohistochemistry for CD14 in alveolar bone (Left panel, ×100; Right panel, ×400). (**F**) Quantification of CD14 protein expression. Error bars represent SD. * *p* < 0.05, *** *p* < 0.001.

**Figure 7 cells-11-02729-f007:**
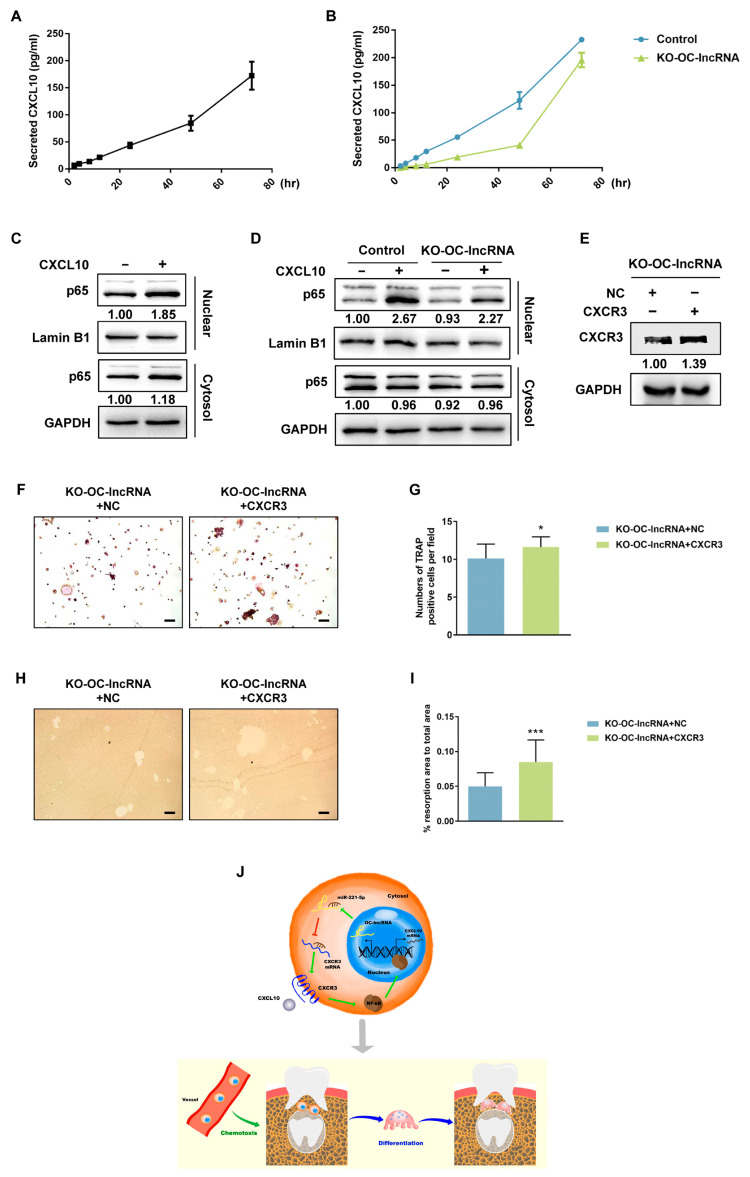
CXCL10–CXCR3 promoted osteoclast differentiation and bone resorption by activating the nuclear translocation of NF-κB p65. (**A**) CXCL10 expression in the supernatants of RANKL-induced RAW 264.7 cells was detected by ELISA. (**B**) CXCL10 expression in the supernatants of RANKL-induced KO-OC-lncRNA RAW 264.7 cells and control. (**C**) CXCL10 stimulation promoted the nuclear translocation of p65 in RAW 264.7 cells. (**D**) OC-lncRNA knock-out impeded the nuclear translocation of p65. (**E**) CXCR3 overexpression in KO-OC-lncRNA RAW 264.7 cells. (**F**) CXCR3 overexpression partially rescued the osteoclast differentiation of KO-OC-lncRNA RAW 264.7 cells. (**G**) Osteoclast counting. (**H**) CXCR3 overexpression partially rescued the bone-resorbing function of KO-OC-lncRNA RAW 264.7 cells. (**I**) Quantification of resorption pits. (**J**) Schematic representation: OC-lncRNA sponges miR-221-5p to increase CXCR3 expression. Error bars represent SD. * *p* < 0.05, *** *p* < 0.001.

**Table 1 cells-11-02729-t001:** Clinical features of 10 CCD patients.

Patient	Sex	Age (Years)	Cranial Sign	Clavicular Sign	Delayed Eruption of Permanent Teeth (Number)	Supernumerary Teeth	Mutation
#1	Male	25	Yes	Yes	Yes (16)	Yes	c.644delG
#2	Female	26	Yes	Yes	Yes (19)	No	c.674G > T
#3	Female	26	Yes	Yes	Yes (7)	Yes	c.559C > T
#4	Male	29	Yes	Yes	Yes (18)	Yes	c.569G > A
#5	Male	20	Yes	Yes	Yes (14)	Yes	c. 514delT ^a^
#6	Female	23	Yes	Yes	Yes (14)	No	—
#7	Male	16	Yes	Yes	Yes (25)	Yes	c.673C > T
#8	Female	18	Yes	Yes	Yes (8)	Yes	c.199C > T
#9	Female	46	Yes	Yes	Yes (7)	Yes	c.557G > C
#10	Male	20	Yes	Yes	Yes (6)	Yes	c.338T > G

^a^ The point mutation selected in RNA-seq.

## Data Availability

The datasets used and/or analyzed during the current study are available from the corresponding author on reasonable request.

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
