# Peer review of "A Novel lncRNA Mediates the Delayed Tooth Eruption of Cleidocranial Dysplasia"

_cells, 2022, doi:10.3390/cells11172729_

Round 1

Reviewer 1 Report

In the present study the authors have identified a novel osteoclast-specific IncRNA that positively regulates osteoclastogenesis and bone resorption via OC-IncRNA-mir221-5p-CXCR3 axis. This novel finding is aimed to provide better understanding of the etiology of tooth impaction in CCD, and the precise molecular mechanism of osteoclastogenesis and bone resorption, aiding the development of effective CCD therapy. While this is an excellent mechanistic study, the authors failed to explain the following points in the manuscript: 

  1. The reason for conducting the mechanistic study on murine cell line instead of human cell line (RAW 264.7). 
  2. The reason for using tissue samples from male mice. Why not female mice were used, irrespective of the fact that the human patients were 50% male and 50% female? 
  3. Which type of tissues were used for RNA and PCR? 
  4. The bar graphs do not include individual data. Therefore, it is requested to include dot plots instead of bar graphs to better understand the distribution of data. 

Overall, a systematic mechanistic study to delineate the mechanism of delayed tooth eruption of cleidocranial dysplasia. 

Author Response

Dear reviewer:

Thank you for your comments concerning our manuscript entitled “A novel lncRNA mediates the delayed tooth eruption of cleidocranial dysplasia” (cells-1886260). Those comments are all valuable and very helpful for revising and improving our paper, as well as the important guiding significance to our research. We have studied comments carefully and have made a correction.

  1. The reason for conducting the mechanistic study on murine cell line instead of human cell line (RAW 264.7).

Response: Function of RUNX2 gene is conserved between human and mouse genomes. RAW 264.7 cells are extensively used in the studies of osteoclasts. RAW 264.7 cells are easy to culture and passage and have better homogeneity. Importantly, mouse model is commonly used to verify the mechanism in vivo. We use mouse cell line to do such research just for test the novel lncRNA function in vivo in future. Take above factors into consideration, RAW 264.7 cells were used in the present study.

  1. The reason for using tissue samples from male mice. Why not female mice were used, irrespective of the fact that the human patients were 50% male and 50% female?

Response: Cleidocranial dysplasia is an autosomal dominant hereditary disease with an incidence of 1/1000,000 and no apparent gender predilection [1]. Since female mice are more affected by osteoporosis and hormone, we prefer male mice in the study of bone-related issues.

  1. Which type of tissues were used for RNA and PCR?

Response: Spleen, colon, skeletal muscle, bladder, kidney, lung, stomach, small intestine, pancreas, liver, brain, heart, and long bone of mice were used for RNA and PCR. Details were added in the Materials and Methods and highlighted.

  1. The bar graphs do not include individual data. Therefore, it is requested to include dot plots instead of bar graphs to better understand the distribution of data.

Response: We have tried preparing figures with dot plots. However, it is not as concise as we expected. Therefore, we prefer bar graphs in our manuscript to comply with the journal style. We also provide the dot plot figure version in the revised Supplementary Materials.

References:

  1. Farronato, G.; Maspero, C.; Farronato, D.; Gioventu, S. Orthodontic treatment in a patient with cleidocranial dysostosis. Angle Orthod 2009, 79, 178-185, doi:10.2319/111307-393.1.

Reviewer 2 Report

The manuscript is generally well written. However, it will be better if the following minor corrections are also made.

Introduction

State the null hypothesis of the study clearly at the end of the introduction.

Materials and methods

Clearly state the inclusion/exclusion criteria used to identify the 10 participants. (on lines 75-78)

The methodology is well established in the materials method section. Many detailed methods have been used.

Discussion

Discuss the null hypothesis and its consequences.

Please indicate the weaknesses and limitations of the study.

References

References 6-7-8-13-14-25 are very old. Please replace with new ones.

Author Response

Dear reviewer:

Thank you for your comments concerning our manuscript entitled “A novel lncRNA mediates the delayed tooth eruption of cleidocranial dysplasia” (cells-1886260). Those comments are all valuable and very helpful for revising and improving our paper, as well as the important guiding significance to our research. We have studied comments carefully and have made a correction.

  1. Introduction

State the null hypothesis of the study clearly at the end of the introduction.

Response: The null hypothesis of this study is indicated in the Introduction part (lines 68-71) which is also paste below.

“In this study, we examined the hypothesis that, OC-lncRNA regulated osteoclastogenesis and bone resorption by the OC-lncRNA–microRNA (miR)-221-5p–CXC chemokine receptor type 3 (CXCR3) axis.”

  1. Materials and methods

Clearly state the inclusion/exclusion criteria used to identify the 10 participants. (on lines 75-78)

The methodology is well established in the materials method section. Many detailed methods have been used.

Response: The diagnostic criteria of cleidocranial dysplasia is added in the Materials and methods (lines 76-83) which is also paste below.

“The diagnostic criteria refer to Mundlos’s report (Clinical features: Brachycephaly; frontal and parietal bossing; open sutures and fontanelleles; delayed closure of fontanelleles; relative prognathism; soft skull in infancy; depressed nasal bridge; hypertelorism; ability to bring shoulders together; narrow, sloping shoulders;  scoliosis; kyphosis; brachydactyly; tapering of fingers; nail dysplasia/hypoplasia; short, broad thumbs; clinodactyly of 5th finger; normal deciduous dentition; supernumerary teeth; impacted, supernumerary teeth; delayed eruption; crowding; malocclusion.) [1].”

  1. Discussion

Discuss the null hypothesis and its consequences.

Please indicate the weaknesses and limitations of the study.

Response: The null hypothesis is discussed in the Discussion part and highlighted, which is also paste below.

“We also identified a novel osteoclast-specific lncRNA that regulates early-stage osteoclastogenesis and bone resorption via the OC-lncRNA–miR-221-5p–CXCR3 axis, which confirmed our initial hypothesis.

These findings are consist with the initial hypothesis that, OC-lncRNA regulated osteo-clastogenesis and bone resorption.”

The weaknesses and limitations of the study (as below) has been added in the discussion part (lines 566-569).

“Further investigations are required to develop reliable analytical methods for demonstrating the function of OC-lncRNA in vivo. A transgenic mouse model with OC-lncRNA knockout is helpful in proceeding with in vivo experiments. Meanwhile, more cleidocranial dysplasia patients should be collected to further verify our findings.”

  1. References

References 6-7-8-13-14-25 are very old. Please replace with new ones.

Response: References 7, 8, 13, and 14 have been replaced by new references as below and highlighted in the revision. While References 6 and 25 are classic dog experiments, which provide better evidence.

Old references

New references

Hitchin, A.D.; Fairley, J.M. Dental management in cleido-cranial dysostosis. Br J Oral Surg 1974, 12, 46-55, doi:10.1016/0007-117x(74)90060-2.

Dorotheou, D.; Gkantidis, N.; Karamolegkou, M.; Kalyvas, D.; Kiliaridis, S.; Kitraki, E. Tooth eruption: altered gene expression in the dental follicle of patients with cleidocranial dysplasia. Orthod Craniofac Res 2013, 16, 20-27, doi:10.1111/ocr.12000.

Migliorisi, J.A.; Blenkinsopp, P.T. Oral surgical management of cleidocranial dysostosis. Br J Oral Surg 1980, 18, 212-220, doi:10.1016/0007-117x(80)90065-7

Chacon, G.E.; Ugalde, C.M.; Jabero, M.F. Genetic disorders and bone affecting the craniofacial skeleton. Oral Maxillofac Surg Clin North Am 2007, 19, 467-474, v, doi:10.1016/j.coms.2007.08.001.

Jensen, B.L.; Kreiborg, S. Development of the dentition in cleidocranial dysplasia. J Oral Pathol Med 1990, 19, 89-93,

620 doi:10.1111/j.1600-0714.1990.tb00803.x

Proffit, W.R.; Frazier-Bowers, S.A. Mechanism and control of tooth eruption: overview and clinical implications. Orthod Craniofac Res 2009, 12, 59-66, doi:10.1111/j.1601-6343.2009.01438.x.

Otto, F.; Thornell, A.P.; Crompton, T.; Denzel, A.; Gilmour, K.C.; Rosewell, I.R.; Stamp, G.W.; Beddington, R.S.; Mundlos,S.; Olsen, B.R., et al. Cbfa1, a candidate gene for cleidocranial dysplasia syndrome, is essential for osteoblast differentiation and bone development. Cell 1997, 89, 765-771, doi:10.1016/s0092-8674(00)80259-7.

Takarada, T.; Nakazato, R.; Tsuchikane, A.; Fujikawa, K.; Iezaki, T.; Yoneda, Y.; Hinoi, E. Genetic analysis of Runx2 function during intramembranous ossification. Development 2016, 143, 211-218, doi:10.1242/dev.128793.

References:

  1. Mundlos, S. Cleidocranial dysplasia: clinical and molecular genetics. J Med Genet 1999, 36, 177-182.

Reviewer 3 Report

Cleidocranial dysplasia is a condition characterized by delayed eruption of permanent teeth among other symptoms. This study focuses on describing a novel OC-lncRNA as a mediator in osteoclast dysfunction in the process of delayed eruption of tooth. The study is presented in detailed manner and provides foundation for further research in this aspect. 

Author Response

Dear reviewer:

Thank you for your comments concerning our manuscript entitled “A novel lncRNA mediates the delayed tooth eruption of cleidocranial dysplasia” (cells-1886260). We appreciate your affirmation of the execution and results of our work. We will give more excitement and enthusiasm to the future work.